# Unraveling the Evolutionary Patterns of Genus *Frontonia*: An Integrative Approach with Morphological and Molecular Data

**DOI:** 10.3390/biology14030289

**Published:** 2025-03-13

**Authors:** Ratih Kusuma Wardani, Ragib Ahsan, Mann Kyoon Shin

**Affiliations:** 1Department of Biological Science, University of Ulsan, Ulsan 44610, Republic of Korea; kusumaratih89@gmail.com (R.K.W.); ragibahsan.jnu@gmail.com (R.A.); 2Program in Organismic and Evolutionary Biology, University of Massachusetts Amherst, Amherst, MA 01003, USA; 3Department of Biology Sciences, Smith College, Northampton, MA 01063, USA

**Keywords:** *Frontonia*, evolutionary history, morphological evolution, diversification analysis, SSU rRNA gene, molecular dating, ancestral state reconstruction

## Abstract

Ciliates, particularly the genus *Frontonia*, have been studied to understand their evolutionary history, but challenges remain. This study investigated the evolutionary patterns of *Frontonia* using genetic and morphological data. Molecular analysis of the SSU rRNA gene revealed four major phylogenetic groups within *Frontonia*, suggesting its paraphyly. The common ancestor existed approximately 420 million years ago, with distinct groups emerging during the Mesozoic era. Diversification analysis showed higher extinction rates than speciation rates within the genus. Morphological traits, including habitat adaptations, were examined through ancestral state reconstructions, revealing a complex evolutionary history. Habitat transitions were not directly linked to morphological traits such as contractile vacuoles, emphasizing the role of genetic diversity and environmental adaptation. These findings provide valuable insights into the interplay between evolution, extinction, and morphology in ciliates, advancing our understanding of biodiversity and evolutionary biology.

## 1. Introduction

The research on ciliates, with a particular focus on the genus *Frontonia*, has indeed advanced significantly due to the integration of molecular and morphological studies. The genus *Frontonia* established nearly two centuries ago, represents a diverse group of ciliate species. The genus *Frontonia* comprises free-living ciliates found in freshwater, marine, and brackish environments, characterized by an elongated body, dense ciliation, and a prominent oral apparatus. As heterotrophs, they play a crucial role in microbial food webs and have a close phylogenetic relationship with *Paramecium* and *Apofrontonia*, making them valuable for studies on ciliate evolution, physiology, and ecological dynamics [1,2,3,4,5,6,7].

While various research efforts have contributed to the understanding of their morphological traits, the phylogenetic relationships and evolutionary history of many ciliate lineages, including *Frontonia*, remain unresolved [8]. Phylogenetic analyses indicate that *Frontonia* is paraphyletic, forming distinct clades with some species grouping with genera like *Apofrontonia*, *Paramecium*, *Stokesia*, and *Marituja*. This challenges its traditional classification and suggests a need for taxonomic revision. However, the possibility of *Frontonia* being monophyletic cannot be entirely dismissed [9].

Molecular analyses, particularly of the SSU rRNA gene, have provided crucial insights into the phylogenetic relationships within *Frontonia*. These analyses have revealed the paraphyletic nature of the genus, with several distinct clades identified. Furthermore, certain species consistently cluster with different genera in the SSU rRNA gene phylogenetic trees, suggesting a complex evolutionary history [8,9,10,11,12].

However, the molecular analysis of the SSU rRNA gene has been the primary tool used to construct phylogenetic trees within *Frontonia*, with limited integration of morphological data beyond species descriptions. To address this gap, a multifaceted approach is proposed in the current study. This approach involves the integration of molecular data, in particular the SSU rRNA gene, for molecular clock analysis. At the same time, morphological data will be used to reconstruct ancestral states, allowing for a more comprehensive understanding of the phylogenetic relationships and evolutionary history of *Frontonia*. By combining molecular and morphological data, the study aims to elucidate the evolutionary patterns, phylogenetic relationships, and divergence time of the genus *Frontonia*. These integrated approaches will provide deeper insights into the diversification patterns and evolutionary processes of the lineages in *Frontonia*.

## 2. Materials and Methods

### 2.1. Sample Collection and Morphological Study

Newly sequenced species of *Frontonia* were collected from different locations in the Republic of Korea. Detailed information on the collection sites can be found in Table 1. The *Frontonia* species were cultivated in petri dishes using water collected from their respective habitats. Initial in vivo cell observations were made using both bright field and differential interference contrast microscopy techniques. Subsequent detailed examinations were performed using a compound microscope, with magnifications ranging from 100× to 1000× for both live cells and stained samples. The Protargol-impregnated method was employed to visualize the structure of the buccal area, the oral apparatus, and the ciliary pattern [13].

### 2.2. DNA Extraction, Amplification, and Sequencing

The genomic DNA extraction was carried out using the RED Extract-N-Amp Tissue PCR Kit from Sigma (St. Louis, MO, USA) according to the manufacturer’s instructions. The polymerase chain reaction (PCR) was performed using the forward primer EUKA (5′-GAC CGT CCT AGT TGG TC-3′) [14] or 82F (5′-CTC GGT AAG CGT CAA AG-3′) [9] and the reverse primers D1, D2 rev2 (5′-GAC TGC ACG TTT AGC TAG CA-3′) or D1, D2 rev4 (5′-GTG CCT GGT TCY TCA GAT TG-3′). PCR amplifications were performed using the TaKaRa ExTaq DNA polymerase kit from TaKaRa Bio-medicals (Otsu, Japan) following a specific protocol: an initial denaturation cycle at 94 °C for 2 min, followed by 37 cycles of denaturation at 95 °C for 30 s, annealing at 50 °C for 40 s, and extension at 72 °C for 4 min. A final extension step was then carried out at 72 °C for 10 min [15].

### 2.3. Alignment and Phylogenetic Analysis

In this study, 71 sequences from the species of the genus *Frontonia* obtained from NCBI and newly sequenced in Republic of Korea (Table A1), 20 sequences from related genera (*Paranassula*, *Paramecium*, *Apofrontonia*, *Stokesia*, *Disematosoma*, *Marituja*, *Lembadion*, *Urocentrum*, and *Tetrahymena*) were selected as reference sequences and two species from the genus *Tetrahymena* were selected as outgroup (*T. pyriformis* & *T. rostrata*) (Figure 1). The alignment for this data set was performed by using MAFFT version 7.0 [16]. Subsequent refinement and masking of the alignments were carried out using G-blocks version 0.91b [17]. To determine the best evolutionary models, jModeltest version 2.1.7 [18,19] was employed. Further analysis included maximum likelihood (ML) using PhyML 3.0 software with 1000 non-parametric bootstrap replicates [20]. In addition, Bayesian Inference (BI) analysis was performed using MrBayes version 3.1 [21] with the best parameter from Jmodeltest result; best model GTR + I + G, p-inv 0.3980, and gamma shape 0.4040.

### 2.4. Molecular Dating and Diversification Analysis

The analysis used 33 sequences from representative species of the genus *Frontonia*, selected on the basis of the availability of both molecular and morphological information (Table A1). In addition, 18 sequences from species of related genera were included in the analysis. Divergence times were estimated using a Bayesian framework that was implemented in BEAST ver. 2.4.5 [22,23]. The software BEAUti ver. 2.4.5 [23] was used to generate the BEAST software input XML file with the following settings; (i) calibrated Yule model, (ii) GTR + I (=0.389) + Γ (=0.412), (iii) four gamma categories for substitution rate heterogeneity, (iv) strict molecular clock, (v) clock rate prior assuming a normal distribution with a mean of 3.88 × 10^−4^ nucleotide substitutions per site per one million years and a 95% credibility interval ranging from 1.24 × 10^−4^ to 9.14 × 10^−4^ [24] and (vi) Yule birth rate set with shape parameter set to 0.001 and scale parameter set to 1000 (gamma shape parameter).

For the estimation of the divergence time, calibration nodes from the genera *Paramecium* and *Tetrahymena* were used. These nodes were based on microfossil evidence from *Paramecium triassicum* and *Tetrahymena rostrata*-like species found in an amber Triassic slab during the Upper Triassic period, approximately 220–230 million years ago (Mya) [25]. The mean divergence time for these calibration nodes was estimated to be 225 Mya. Markov Chain Monte Carlo (MCMC) analyses started with a random seed and ran for 6,000,000 generations, with trees and all other parameters saved every 10,000th iteration. The quality of the MCMC analysis was assessed using Tracer version 1.6 [26] to ensure convergence and adequate burn-in. The final maximum credibility tree was generated using TreeAnnotator ver. 1.8.1 [22] after discarding the first 10% of sampled trees.

To analyze the diversification dynamics of *Frontonia*, we used DivBayes and SubT [27], two Bayesian-based approaches for estimating speciation and extinction rates. DivBayes infers diversification rates by integrating ancestral divergence times with the observed and predicted species numbers, assuming a birth-death process. The method utilizes Bayesian posterior distributions to estimate net diversification, accounting for rate variation over time. SubT estimates speciation rates using 95% height data from chronograms generated by BEAST2, incorporating ultrametric tree node depths. To correct for taxon sampling biases, we included predicted species numbers where actual data were unavailable (Table A2), improving accuracy in rate estimation. These methods were chosen due to their robustness in handling incomplete taxon sampling and their ability to infer diversification trends over evolutionary timescales.

### 2.5. Reconstruction of Ancestral Morphologies

For the prediction of ancestral states, 33 species of the genus *Frontonia* and *Apofrontonia dohrni* were selected as representative taxa. The selection was based on the availability and credibility of their molecular and morphological data, as shown in Table A2. To perform the ancestral state reconstruction, 12 important and significant characters were selected within the genus *Frontonia*. These characters included the number of ciliary rows in peroral membranes (PM), the number of vestibular kineties (VK), the number of postoral kineties (PK), the number of ciliary rows in peniculi 1–3, the structure of P3 (shortened and linearity with P1 and P2), the habitat, the number of contractile vacuoles (CV), the number of contractile vacuolar pores (CVP) and the presence of contractile canals (CC). The character matrix and ancestral state reconstruction were performed using the parsimony model in Mesquite software ver. 3.70 [28]. The reconstruction of ancestral states was based on the topology of the best likelihood tree obtained from the RAxML analysis performed on CIPRES [29].

## 3. Results

### 3.1. Phylogenetic Analyses

The phylogenetic analyses based on SSU rRNA gene sequences have provided insights into the evolutionary relationships within the genus *Frontonia*, revealing the existence of four main groups: Group I: This group includes species such as *F. canadensis*, *F. subtropica*, *F. salmastra*, *F. sinica*, *F. magna*, *F. mengi*, *F. tchibisovae*, and *F. lynni*. Both Bayesian Inference (BI) and Maximum Likelihood (ML) analyses strongly support this group, with node support values of 1.00/100%. Group II: Comprising species such as *F. paramagna*, *F. leucas*, *F. vesiculosa*, *F. vernalis*, *F. shii*, *F. paravernalis*, *F. leucas*, *F. angusta*, and *F*. cf. *leucas*, Group II also shows robust node support values of 1.00/100% for both BI and ML. Group III: This group includes species such as *F. didieri*, *F. ocularis*, *F. elegans*, *F. pusilla*, and *F. anatolica*, with slightly lower node support values of 0.98/85% for BI and ML compared to Groups I and II. Group IV: Including species such as *F. terricola*, *F*. cf. *acuminata*, *F. acuminata*, *F. atra*, *F. apoacuminata*, *F. minuta*, and *F*. cf. *atra*, Group IV shows moderate node support (BI/ML, 1.00/88%) (Figure 1).

Furthermore, Group III shows a close relationship with the genus *Apofrontonia*, albeit with lower node support (BI/ML, 0.81/60%), and together this sister clade formed a clade with the genus *Paramecium* with full support. Group IV forms a cluster with the genera *Stokesia*, *Marituja*, and *Disematostoma*, with strong node support (BI/ML, 1.00/100%). These phylogenetic relationships shed light on the evolutionary dynamics and diversification patterns within the genus *Frontonia* highlighting both strong and moderate support for the identified groups and their relationships with related taxa (Figure 1).

### 3.2. Estimation of Divergence Times and Diversification Using the SSU rRNA Gene

The divergence time within the genus *Frontonia* can be estimated using two methods: relaxed clock and strict clock. A relaxed clock is commonly used at higher taxonomic levels such as family or order, while a strict clock is favored for intraspecific level analyses where low rates of variation between branches are expected [30,31,32,33]. In the case of the genus *Frontonia*, the strict clock approach is more suitable for estimating divergence time due to the limited data set within the genus level, and the minimal variation between branches.

The analyses of divergence time in the genus *Frontonia* performed in this study show that Peniculia is estimated to have originated approximately 750 million years ago (Mya), confirming the previous findings of Rataj and Vďačný (2018) [24] (Figure 2). Using the strict clock approach, the SSU rRNA gene showed a mean clock rate of 2.36 × 10^−4^ per year. Node calibrations for the *Tetrahymena* clade and *Paramecium* clades indicate their emergence at 223 Mya and 226 Mya, respectively. The common ancestor of the genus *Frontonia* appeared around 420 Mya and serves as an ancestral point for all members of the Penicullida. Notably, the emergence of three clades within the genus *Frontonia* occurred relatively recently: Group I around 172 Mya, Group II around 83 Mya, Group III around 115 Mya, and Group IV around 190 Mya (Figure 2).

The results obtained from Divbayes and SubT for speciation and extinction indicate a higher extinction rate compared to the speciation rate with genus *Frontonia*: 0.826 species/year over speciation rate: 0.011 species/year (DivBayes 1.1) and extinction rate: 0.613 species/year over speciation rate: 0.016 species/year (SubT1.1).

### 3.3. Reconstruction of Ancestral Character State

The analysis of ancestral states in the genus *Frontonia* reveals intriguing evolutionary patterns across several morphological characters;

Vestibular kineties (VK): Group III retains the ancestral state of three rows of VK, whereas Group I and II show increases to four and five rows, respectively. Notably, *F. canadensis* shows transitions from three to five and back to three rows, consistent with the ancestral state of the genus *Frontonia*. Group IV undergoes significant changes, expanding to four, five, and even six rows (Figure 3A).Postoral kineties (PK): Group I shows various increases in PK number (six to eight rows), Group II retains the ancestral state PK number except for *F. magna*, Group III shows a decrease in PK number except for *F, anatolica*, whereas Group IV shows both decreasing and increasing tendencies, with some species reaching up to six rows (Figure 3B).Peroral membrane (PM): Groups II and III show an increase in PM ciliary rows, although species such as *F. sinica*, *F. salmastra*, and *F. lynni* revert from two rows to one (Figure 3C).Peniculi 1 and 2 (P1 and P2): Group I shows a marked increase from four to five rows, while other groups, with a few exceptions, retain the ancestral state (Figure 3D,E).Peniculi 3 evolution: Groups I and II show an increase in ciliary rows from three to four or five, whereas *F. mengi* and *F. sinica* uniquely decrease to two rows. Group IV members retain the ancestral state with several species increasing to four or five rows, while Group III decreases from the three ancestral ciliary rows to two. Most *Frontonia* species retain the ancestral linear structure of peniculi 3, with a shortening of peniculi 3 considered an ancestral character (Figure 3F–H).Habitat adaptation: The genus *Frontonia* originates from brackish habitats before adapting to freshwater environments. Group III remains in the brackish environment and adapts to marine habitats for several species, while Group I and IV retain a freshwater adaptation, except for *F. acuminata*, which adapts to a marine environment. Group II evolves from freshwater to marine and brackish habitats (Figure 4).Contractile vacuole (CV): The ancestral analysis suggests that the genus originally possessed a single CV, a character often retained. Group III, excluding some species, increases from one to two CVs, while unique species in Group IV show up to 10 CVs per individual (Figure 4).Contractile vacuolar pores (CVP) and collecting canals (CC): The evolution of CVPs and CCs is not closely correlated with the evolution of CVs (Figure 4).

These results shed light on the evolutionary history and adaptive strategies of *Frontonia* species, providing valuable insights into morphological evolution and habitat preferences.

## 4. Discussion

### 4.1. Phylogenetic Relationships and Morphological Distinctions in Frontonia

In this study, a paraphyletic phylogenetic tree was obtained, aligning with previous findings. While Groups III and IV are consistently separated from Groups I and II, it remains premature to conclude that they do not belong to the genus *Frontonia*. Morphological analysis reveals that both groups share multiple synapomorphic traits with Groups I and II, suggesting that redefining them as distinct genera would require a comprehensive taxonomic revision of *Frontonia*.

Group III and the genus *Apofrontonia* form a sister clade closely related to the genus *Paramecium* clade. However, their phylogenetic proximity is not mirrored in their morphological characteristics. For instance, *Frontonia* Group III possesses a peniculi 3 structure similar to peniculi 1 and 2, whereas genus *Apofrontonia* exhibits a distinct structure, and the genus *Paramecium* features quadrulus instead of peniculi 3. Likewise, *Frontonia* Group IV and its sister clade, which includes the genera *Marituja* and *Stokesia*, exhibit notable differences in peniculi 3 morphology, further distinguishing them from peniculi I and II. These findings underscore the complex interplay between phylogenetic relationships and morphological evolution within *Frontonia* [34].

### 4.2. Evolutionary Patterns of Morphological Characters in the Genus Frontonia

The evolution of morphological traits in *Frontonia* reflects both stable characteristics and dynamic modifications, influenced by selective pressures and developmental constraints. Key traits of the oral apparatus—including the number of vestibular kineties (VK), postoral kineties (PK), peroral membrane (PM) ciliary rows, and peniculi structures—exhibit distinct evolutionary trajectories [8]

#### 4.2.1. Oral Apparatus Evolution: Patterns and Constraints

Vestibular kineties show a general increasing trend in *Frontonia*, suggesting a functional or adaptive advantage. However, occasional reversals, such as in *F. canadensis*, imply a potential evolutionary limit, possibly due to structural constraints or trade-offs in ciliary coordination. Meanwhile, postoral kineties demonstrate a more variable pattern, reflecting relaxed selection or species-specific functional adaptations.

The peroral membrane exhibits both doubling trends and reversals within Group II and III members, highlighting its plasticity. The reversion to a single-rowed PM in *F. sinica* and related species suggests that the ancestral state remains functionally viable, possibly due to energetic efficiency or redundancy in feeding structures. This aligns with the concept that trait reversibility may be constrained by ecological pressures and developmental flexibility.

Peniculi structures further emphasize differential evolutionary stability. Peniculi 1 and 2 remain conserved across species, whereas peniculi 3 undergoes frequent modifications in number, length, and structure. This suggests that while peniculi 1 and 2 perform essential functions limiting their variability, peniculi 3 may provide additional structural flexibility, allowing species-specific adaptations.

#### 4.2.2. Contractile Vacuole Evolution: Stability vs. Plasticity

The contractile vacuole (CV) system, responsible for osmoregulation, is often shaped by habitat conditions in cilates such as *Frontonia* [35,36,37,38]. However, ancestral state reconstruction indicates no strict correlation between habitat and CV number, nor its associated traits (contractile vacuolar pores and contraction canals). This suggests that osmotic regulation in *Frontonia* is governed by factors beyond direct environmental influence, such as cellular physiology, metabolic constraints, or phylogenetic inheritance.

Interestingly, while Group II members maintain a stable CV number despite habitat shifts, Group III exhibits an evolutionary transition from a single CV to two CVs. This divergence suggests that CV evolution may be lineage-specific rather than driven by direct ecological pressures, supporting the idea that some morphological features persist due to historical contingency rather than continuous selective pressure.

#### 4.2.3. Reversals and the Neutral Morphological Theory

The presence of reversals in both peroral membrane structure and vestibular kineties aligns with the neutral morphological theory, which proposes that morphological adaptations evolve within a limited range dictated by functional efficiency rather than continuous directional selection. Similar reversions in ciliary structures across different *Frontonia* lineages reinforce the idea that some morphologies represent optimal configurations, leading to recurrent evolutionary patterns.

While some lineages exhibit long-term stability in traits, others undergo modifications, suggesting a balance between conserved developmental pathways and selective pressures favoring structural plasticity. Additional studies on genetic regulation and functional morphology could provide further insights into the mechanisms driving these evolutionary patterns [39].

### 4.3. Evolutionary History of Genus Frontonia

The evolutionary history of the genus *Frontonia* provides insights into its emergence and divergence over millions of years and sheds light on the environmental factors and evolutionary processes that have shaped its diversity. The common ancestor of the four groups within the genus *Frontonia* and members of Penicullida emerged and diverged approximately 420 million years ago during the Palaeozoic era (Figure 2) [40]. Each group or clade within *Frontonia* originated at different geological times. Groups I and II shared a common ancestor at the beginning of the Mesozoic era about 230 million years ago, with Group I diverging about 172 million years ago and Group II diverging about 83 million years ago. Group III, morphologically close to the genus *Apofrontonia*, shared a common ancestor at the beginning of the Mesozoic era, about 185 million years ago, while Group III itself diverged about 115 million years ago. Group IV appeared about 190 million years ago, making it one of the oldest groups within *Frontonia*. Group I is the youngest, emerging at the end of the Mesozoic era.

#### 4.3.1. Paleozoic Environmental Influences on *Frontonia* Evolution

The Paleozoic era, encompassing the Cambrian, Ordovician, Silurian, Devonian, and Carboniferous periods, was marked by significant evolutionary events, including the Cambrian explosion—a period of rapid speciation [40]. Chronogram analysis suggests that the ancestor of *Frontonia* likely diverged during this era. The warm climatic conditions of the Cambrian facilitated species diversification; however, fossil records indicate significant environmental fluctuations, driven by glacial and deglacial cycles, which influenced sea levels and temperatures.

Two major mass extinctions in the Paleozoic era, occurring at the end of the Devonian and Permian (late Carboniferous period), were linked to abrupt environmental shifts [41,42]. Although direct fossil evidence for ciliates is lacking, these extreme conditions may have constrained *Frontonia*’s ancestral diversification, as reflected in the chronogram, which shows long branches with minimal speciation events.

Diversification tests suggest a higher extinction rate than the speciation rate in *Frontonia*, indicating the challenges faced by ancestral populations in maintaining their existence over geological time scales. The surviving *Frontonia* lineages persisted into the early Mesozoic, where they underwent diversification alongside other microbial taxa, such as dinoflagellates and algae, as global temperatures stabilized [43,44].

#### 4.3.2. Paraphyly and SSU rRNA Gene Polymorphism in *Frontonia*

The current paraphyly within the genus *Frontonia* may be due to SSU rRNA gene polymorphism in its ancestor, contributing to the high genetic diversity observed in the past. Harsh environmental conditions during the Palaeozoic era probably subjected ancestral *Frontonia* populations to natural selection, resulting in the decline of many populations. However, surviving populations retained variations in their SSU rRNA gene, potentially leading to divergent evolutionary trajectories. Over time, during the Mesozoic era, these divergent populations underwent speciation, possibly in response to changing environmental conditions, as supported by recent studies [8,45].

## 5. Conclusions

Our study provides a comprehensive analysis of the morphological evolution in *Frontonia*, with a focus on the oral apparatus and contractile vacuole characteristics. Ancestral state reconstruction reveals distinct evolutionary patterns, where the number of vestibular kineties tends to increase, while postoral kineties exhibit greater variability. The peroral membrane demonstrates both expansion and reversal, indicating morphological plasticity within certain lineages. Similarly, peniculi 1 and 2 remain evolutionarily stable, whereas peniculi 3 show greater structural modifications.

Although habitat transitions have been considered a key factor influencing ciliate morphology, our findings suggest that the number of contractile vacuoles and related structures remains largely stable despite environmental shifts. This indicates that habitat alone does not drive their evolution, and other selective pressures or genetic constraints may play a more significant role. The presence of both stable and reversible traits within *Frontonia* supports the idea of morphological evolution occurring within a constrained framework, aligning with the neutral morphological theory.

The historical perspective of the genus *Frontonia* traces its origins back to the Cambrian explosion, revealing survival challenges during the Paleozoic era and subsequent diversification throughout the Mesozoic era. The observed paraphyly within *Frontonia* is attributed to SSU rRNA gene polymorphism in its ancestor, reflecting high genetic diversity and adaptation to changing environmental conditions over time.

## Figures and Tables

**Figure 1 biology-14-00289-f001:**
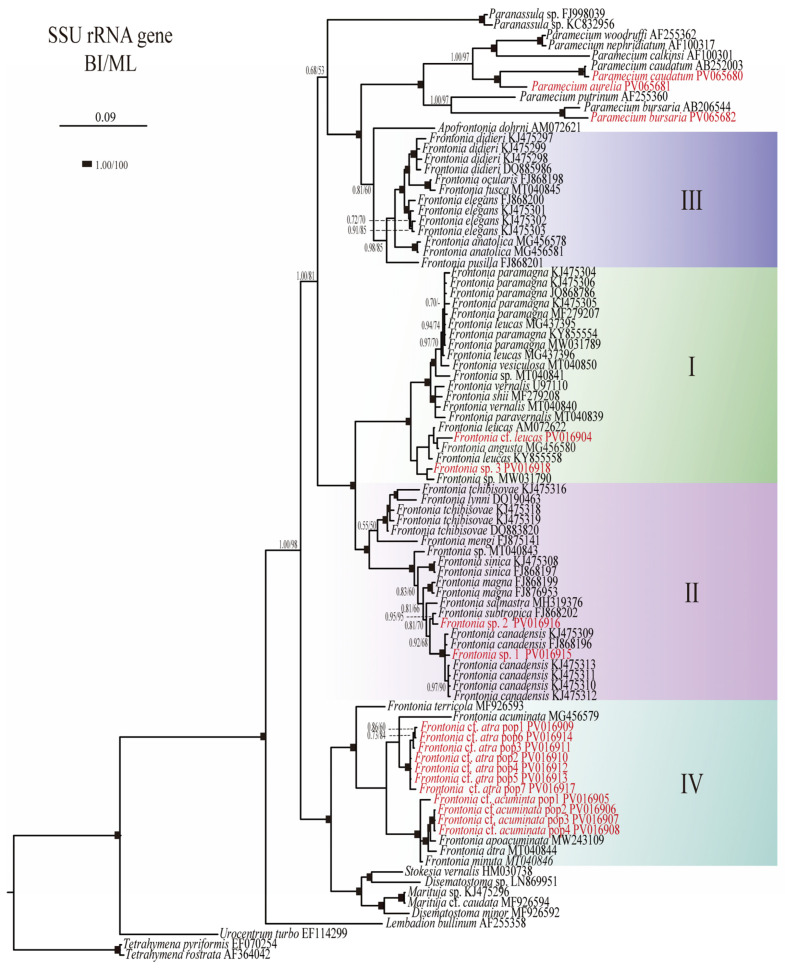
Phylogenetic tree generated from Bayesian Inference (BI) using Mr Bayes ver. 3.1 software and Maximum Likelihood (ML) from PHYML ver. 3.0 based on the SSU rRNA (**top**) and SSU-ITS-LSU rRNA (**bottom**) gene sequences of the genus *Frontonia* and related genera. The scale bar indicates the number of base changes per 1000 nucleotide positions in BI analysis. Node support is represented as follows: BI posterior probability/ML bootstrap. The newly sequenced species of the genus *Frontonia* are shown in red. Each clade or group of genus *Frontonia* in phylogenetic tree represent by I–IV.

**Figure 2 biology-14-00289-f002:**
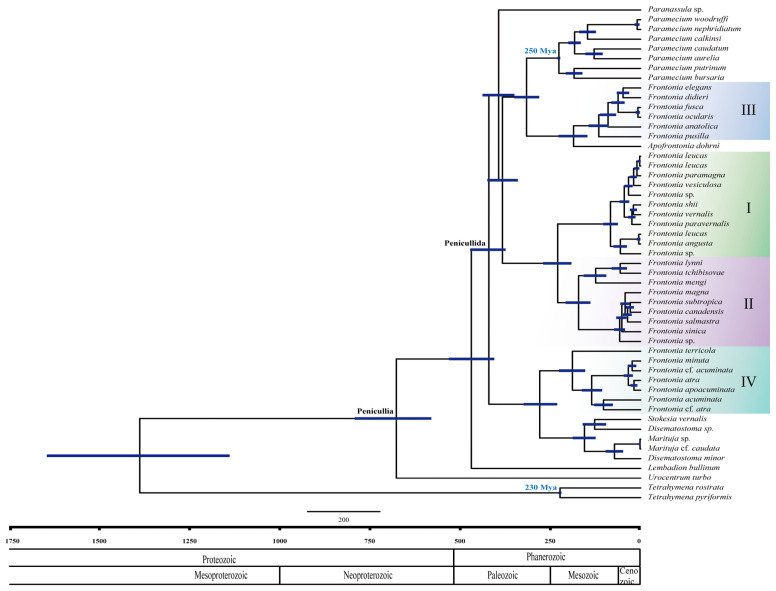
Maximum credibility tree showing posterior means of divergence times of the genus *Frontonia* and related genera obtained using Bayesian relaxed molecular dating in BEAST. The 95% credibility intervals are shown as bars for all nodes. The horizontal axis represents the time scale in millions of years. Each clade or group of genus *Frontonia* in phylogenetic tree represent by I–IV.

**Figure 3 biology-14-00289-f003:**
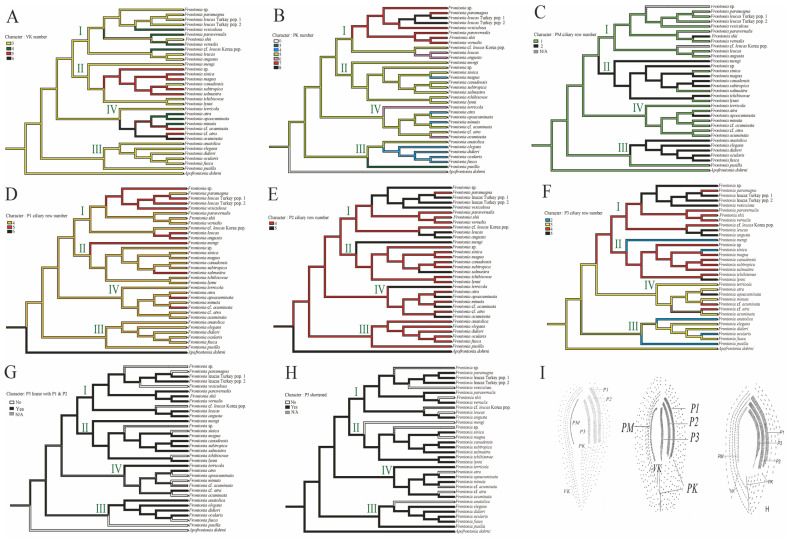
Ancestral state reconstruction of morphological characteristics; (**A**) vestibular kinety (VK) number, (**B**) postoral kinety (PK) number, (**C**) peroral membrane (PM) row number, (**D**) peniculi 1 (P1) row number, (**E**) peniculi 2 (P2) row number, (**F**) peniculi 3 (P3) row number, (**G**) linearity of peniculi 3 (P3) in relation to peniculi 1 (P1) and peniculi 2 (P2), (**H**) length of peniculi 3 (P3) in relation to peniculi 1 (P1) and peniculi 2 (P2) (shortened), (**I**) *F. shii* [7] (**right**) as an example of species in which P3 has the same structure as P1 and P2, *F. paramagna* [7] (**middle**) as an example of species in which P3 does not have the same structure as P1 and P2, and *F*. *canadensis* [5] (**left**) as an example of species in which P3 is shortened compared to P1 and P2. 0: no, 1: yes. Each clade or group of genus *Frontonia* in phylogenetic tree represent by I–IV.

**Figure 4 biology-14-00289-f004:**
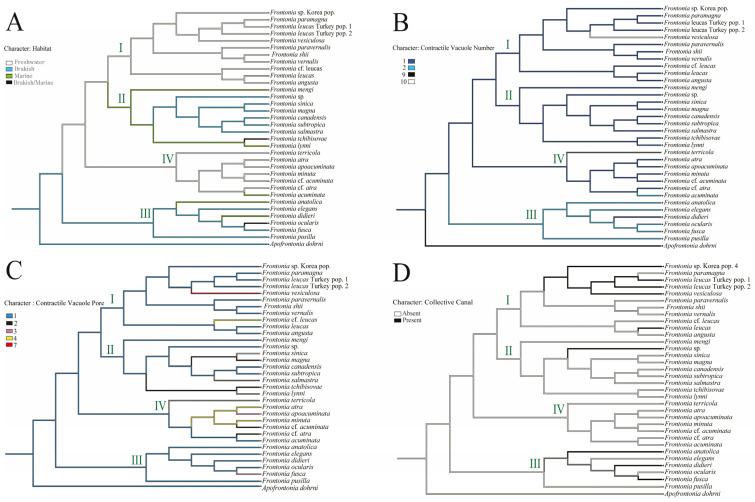
Reconstruction of ancestral state of *Frontonia* habitat (**A**) and morphological characteristics; (**B**) number of contractile vacuole (CV), (**C**) number of contractile vacuolar pore (CVP) and (**D**) presence of contractile canal (CC). Each clade or group of genus *Frontonia* in phylogenetic tree represent by I–IV.

**Table 1 biology-14-00289-t001:** *Frontonia* species from the Republic of Korea and their accession number for SSU rRNA gene sequence.

Taxon	Collection Site	Accession No.
*Frontonia* cf. *acuminata* pop1	Gwangju, Republic of Korea	PV016905
*Frontonia* cf. *acuminata* pop2	Ulsan, Republic of Korea	PV016906
*Frontonia* cf. *acuminata* pop3	Gunsan, Republic of Korea	PV016907
*Frontonia* cf. *acuminata* pop4	Ulsan, Republic of Korea	PV016908
*Frontonia* cf. *atra* pop1	Daejon, Republic of Korea	PV016909
*Frontonia* cf. *atra* pop2	Gunsan, Republic of Korea	PV016910
*Frontonia* cf. *atra* pop3	Masan, Republic of Korea	PV016911
*Frontonia* cf. *atra* pop4	Chungcheongbukdo, Republic of Korea	PV016912
*Frontonia* cf. *atra* pop5	Gyeongsangnamdo, Republic of Korea	PV016913
*Frontonia* cf. *atra* pop6	Ulsan, Republic of Korea	PV016914
*Frontonia* cf. *atra* pop7	Gyeongju, Republic of Korea	PV016917
*Frontonia* cf. *leucas*	Gwangju, Republic of Korea	PV016904
*Frontonia* sp. 1	Ulsan, Republic of Korea	PV016915
*Frontonia* sp. 2	Ulsan, Republic of Korea	PV016916
*Frontonia* sp. 3	Gyeongju, Republic of Korea	PV016918

## Data Availability

The data presented in this study are available on request from the corresponding author.

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
