# Peer review of "Unraveling the Evolutionary Patterns of Genus Frontonia: An Integrative Approach with Morphological and Molecular Data"

_biology, 2025, doi:10.3390/biology14030289_

Round 1
Reviewer 1 Report
Comments and Suggestions for Authors
‘Unravelling the Evolutionary Patterns of Genus Frontonia: An Integrative Approach with Morphological and Molecular Data’ by Wardani, Ahsan, and Shin presents a valuable and interesting analysis of Frontonia. It presents an overview of the genus and its evolutionary history with the inclusion of numerous new species. Several perspectives arise from this analysis that could be treated in a bit more detail.
A species extension rate 38 to 75 time higher than the speciation rate seems remarkable. The means of establishing these rates is treated only briefly and a discussion of these results seems in order. The analysis of physical features such as contractile vacuoles and environmental adaptation is valuable.
The main phylogenetic tree (Figure 1) displays a salient feature that suggest additional comment and analysis. Group I and II clearly form a unified clade. However, group III is much more closely associated with Paramecium. This association should be discussed, possibly group III organisms should be viewed as Paramecium rather than Frontonia. Similarly group IV should be considered for renaming based on their distance from the Frontonia of group I and II.
The suggestion that Frontonia species numbers increased during the Cambrian period followed by mass extinctions in the Paleozoic era is an interesting observation deserving of additional comment.
Minor comments:
Line 75- Protargol-impregnated [trade name]
Line 122- found in on an amber
Line Fig 3 & 4 The colors are hard (impossible) to distinguish
Line Fig 4A Character: Habitat is too small to read
Author Response
(Suggestion) a species extension rate 38 to 75 time higher than the speciation rate seems remarkable. The means of establishing these rates is treated only briefly and a discussion of these results seems in order.
(Answer) Thank you for your suggestion. We have revised the methodological description to clarify how the values were obtained using the DivBayes and SubT software. These modifications can be found on lines 144–154.
(Suggestion) The analysis of physical features such as contractile vacuoles and environmental adaptation is valuable.
(Answer) We have revised the discussion section (lines 365–376) to provide a more detailed explanation of the relationship between environmental adaptation and the evolution of the contractile vacuole.
(Suggestion) The main phylogenetic tree (Figure 1) displays a salient feature that suggest additional comment and analysis. Group I and II clearly form a unified clade. However, group III is much more closely associated with Paramecium. This association should be discussed, possibly group III organisms should be viewed as Paramecium rather than Frontonia. Similarly group IV should be considered for renaming based on their distance from the Frontonia of group I and II.
(Answer) Thank you for your suggestion. In response, we have incorporated a discussion on this topic in the discussion section (line 307-338).
(Suggestion) The suggestion that Frontonia species numbers increased during the Cambrian period followed by mass extinctions in the Paleozoic era is an interesting observation deserving of additional comment.
(Answer) We have modified the discussion part to elaborate Paleozoic era influences to Frontonia evolution (line 408-485)
(Suggestion) Minor comments:
Line 75- Protargol-impregnated [trade name]
(Answer) We used the handmade protargol according to the protocol [1]
Line 122- found in on an amber
(Answer) We thank you for checking typos.
Line Fig 3 & 4 The colors are hard (impossible) to distinguish
(Answer) We used different images to distinguish the colors.
Line Fig 4A Character: Habitat is too small to read
(Answer) We have modified the part according to your suggestion.
References ;
- Pan, X.; Bourland, W.A.; Song, W. Protargol Synthesis: An in-House Protocol. J. Eukaryot. Microbiol. 2013, 60, 609–614, doi:10.1111/jeu.12067.
Reviewer 2 Report
Comments and Suggestions for Authors
This manuscript investigated the evolutionary patterns of Frontonia using genetic and morphological data. From a methodological perspective, the multifaceted approach is a significant advancement. The results revealed paraphyly within Frontonia, identifying four groups that emerged in the Mesozoic era. The authors explain this paraphyly through the SSU rRNA gene polymorphism, which is reasonable. The findings highlight the complex evolutionary patterns of the genus. However, due to the following shortcomings, revisions are necessary.
Main shortcomings:
- In the first paragraph of the Introduction, the authors should describe the genus Frontonia in detail, including its unique morphological, physiological, and ecological characteristics, as well as its research significance.
- In the same first paragraph, the authors mention that phylogenetic relationships remain unresolved. Could you please provide some specific examples?
- Lines 96-106: I found several tools mentioned, such as MAFFT, jModeltest, PhyML and MrBayes, but their versions appear to be older. Why not chose newer versions or alternative software like RAxML?
- Lines 105-106: For Bayesian Inference, it would better to add detailed parameters and a convergent test.
- In the section of “Estimation of divergence times”, I think the result of divergence times is not is uncertain based on a single gene (SSU rRNA).
Author Response
(Suggestion) In the first paragraph of the Introduction, the authors should describe the genus Frontonia in detail, including its unique morphological, physiological, and ecological characteristics, as well as its research significance.
(Answer) We have modified the introduction based on your suggestion (line 49-53 )
(Suggestion) In the same first paragraph, the authors mention that phylogenetic relationships remain unresolved. Could you please provide some specific examples?
(Answer) We have modified the introduction based on your suggestion (line 56-60 )
(Suggestion) Lines 96-106: I found several tools mentioned, such as MAFFT, jModeltest, PhyML and MrBayes, but their versions appear to be older. Why not chose newer versions or alternative software like RAxML?
(Answer) We acknowledge the reviewer’s concern regarding the version of the software used in our analysis. At the time of our study, the version of the software we used was the one available and widely used for phylogenetic reconstruction. Despite subsequent updates, the version we used was sufficient to analyze our dataset and produce the same or reliable results as the newer version. Additionally, we chose PhyML over RAxML because it provided better resolution of clades and improved node support in our dataset. While RAxML is known for its computational efficiency and handling of large datasets, PhyML performed better in terms of resolving phylogenetic relationships within Frontonia, as reflected in higher bootstrap support values and a more stable tree topology. Given these advantages, we considered PhyML to be the more suitable method for our study.
(Suggestion) Lines 105-106: For Bayesian Inference, it would better to add detailed parameters and a convergent test.
(Answer) We have modified the methodology part based on your suggestion (Line 119-120 )
(Suggestion) In the section of “Estimation of divergence times”, I think the result of divergence times is not is uncertain based on a single gene (SSU rRNA).
(Answer) We appreciate the reviewer’s concern regarding the potential uncertainty of divergence time estimation based on a single gene (SSU rRNA). However, SSU rRNA gene has been widely used in previous studies for estimating divergence times in [2–4]. While we acknowledge that single-gene analyses have limitations, SSU rRNA gene remains the best available option given the current data resources. Furthermore, its use ensures comparability with previous studies using the same marker. We agree that the inclusion of additional molecular markers, such as ITS, LSU rRNA, or protein-coding genes (e.g., COI, HSP70), could improve the accuracy of divergence time estimation. However, the availability of such markers for Frontonia in public databases is currently limited. Future studies integrating multi-gene or genome-wide phylogenomic approaches will provide a more comprehensive framework for understanding the evolutionary history of Frontonia.
References;
- Vďačný, P.; Breiner, H.-W.; Yashchenko, V.; Dunthorn, M.; Stoeck, T.; Foissner, W. The Chaos Prevails: Molecular Phylogeny of the Haptoria (Ciliophora, Litostomatea). Protist 2014, 165, 93–111, doi:10.1016/j.protis.2013.11.001.
- Vd’ačný, P.; Rajter, L. Reconciling Morphological and Molecular Classification of Predatory Ciliates: Evolutionary Taxonomy of Dileptids (Ciliophora, Litostomatea, Rhynchostomatia). Mol. Phylogenet. Evol. 2015, 90, 112–128, doi:10.1016/j.ympev.2015.04.023.
- Rajter, Ľ.; Vďačný, P. Rapid Radiation, Gradual Extinction and Parallel Evolution Challenge Generic Classification of Spathidiid Ciliates (Protista, Ciliophora). Zool. Scr. 2016, 45, 200–223, doi:10.1111/zsc.12143.
Reviewer 3 Report
Comments and Suggestions for Authors
The paper represents the results of study the evolution and phylogeny of ciliate genus Frontonia with usage of molecular analysis based on SSU rRNA gene as well as analysis of morphological characters. Four intrageneric groups of species that possible diverged in Palaeozoic era were distinguished based upon molecular data. It was shown that the extinction rates were higher than speciation rates within the genus.
The results of comparative morphological analysis are also interesting but less earnestly. This, for example concerns the reconstruction of the ancestral CV apparatus in Frontonia.
The same is more concern the reconstruction of habitats. At first it is incomprehensible why habitats are morphological characters. At second the latter is based on the analysis of habitats of recent Frontonia species, whereas habitats of ancient species possible significantly differed in previous geological eras.
Perhaps authors should more emphasize the hypothetical nature of their conclusions.
Overall, the work was conducted at a high methodological level and can be recommended for publication.
Author Response
(Suggestion) The paper represents the results of study the evolution and phylogeny of ciliate genus Frontonia with usage of molecular analysis based on SSU rRNA gene as well as analysis of morphological characters. Four intrageneric groups of species that possible diverged in Palaeozoic era were distinguished based upon molecular data. It was shown that the extinction rates were higher than speciation rates within the genus. The results of comparative morphological analysis are also interesting but less earnestly. This, for example concerns the reconstruction of the ancestral CV apparatus in Frontonia.
(Answer) Thank you for your suggestion. We have revised the discussion section to elaborate on the morphological evolution of Frontonia (Line 340–390).
(Suggestion) The same is more concern the reconstruction of habitats. At first it is incomprehensible why habitats are morphological characters. At second the latter is based on the analysis of habitats of recent Frontonia species, whereas habitats of ancient species possible significantly differed in previous geological eras.
(Answer) We acknowledge that habitat itself is not a morphological character, but an important character in Frontonia species identification and delimitation, as supported by previous studies [5]. In our study, we aimed to explore whether habitat shifts influence morphological traits, particularly the contractile vacuole (CV) apparatus, and we incorporated habitat as a character to infer the ancestral habitat of Frontonia based on synapomorphy in the concept of habitat as a character, as habitat is commonly used as a character in evolutionary trajectory analyses of ciliates [5–7]. In addition, we have also made adjustments to the legend of Figure 4 in the response to your concerns.
(Suggestion) Perhaps authors should more emphasize the hypothetical nature of their conclusions.
(Answer) Thank you for your suggestion. I have revised the conclusion section to improve its clarity and objectivity (Line 497-510).
References;
- Zhao, Y.; Yi, Z.; Warren, A.; Song, W.B. Species Delimitation for the Molecular Taxonomy and Ecology of the Widely Distributed Microbial Eukaryote Genus Euplotes (Alveolata, Ciliophora). Proc. R. Soc. B Biol. Sci. 2018, 285, doi:10.1098/rspb.2017.2159.
- Syberg-Olsen, M.J.; Irwin, N.A.T.; Vannini, C.; Erra, F.; Di Giuseppe, G.; Boscaro, V.; Keeling, P.J. Biogeography and Character Evolution of the Ciliate Genus Euplotes (Spirotrichea, Euplotia), with Description of Euplotes Curdsi Sp. Nov. PLoS One 2016, 11, 1–18, doi:10.1371/journal.pone.0165442.
- Sun, P.; Clamp, J.; Xu, D.; Huang, B.; Shin, M.K. An Integrative Approach to Phylogeny Reveals Patterns of Environmental Distribution and Novel Evolutionary Relationships in a Major Group of Ciliates. Sci. Rep. 2016, 6, 1–12, doi:10.1038/srep21695.